# AMPK-Nrf2 Signaling Pathway in Phrenic Motoneurons following Cervical Spinal Cord Injury

**DOI:** 10.3390/antiox11091665

**Published:** 2022-08-26

**Authors:** Pauline Michel-Flutot, Laurie Efthimiadi, Lynda Djerbal, Therese B. Deramaudt, Marcel Bonay, Stéphane Vinit

**Affiliations:** Université Paris-Saclay, UVSQ, Inserm, END-ICAP, 78000 Versailles, France

**Keywords:** Nrf2, spinal cord injury, phrenic motoneuron, AMPK, neuroinflammation, antioxidant, rat

## Abstract

High spinal cord injuries (SCI) induce the deafferentation of phrenic motoneurons, leading to permanent diaphragm paralysis. This involves secondary injury associated with pathologic and inflammatory processes at the site of injury, and at the level of phrenic motoneurons. In the present study, we evaluated the antioxidant response in phrenic motoneurons involving the AMPK-Nrf2 signaling pathway following C2 spinal cord lateral hemi-section in rats. We showed that there is an abrupt reduction in the expression of phosphorylated AMPK and Nrf2 at one hour post-injury in phrenic motoneurons. A rebound is then observed at one day post-injury, reflecting a return to homeostasis condition. In the total spinal cord around phrenic motoneurons, the increase in phosphorylated AMPK and Nrf2 occurred at three days post-injury, showing the differential antioxidant response between phrenic motoneurons and other cell types. Taken together, our results display the implication of the AMPK-Nrf2 signaling pathway in phrenic motoneurons’ response to oxidative stress following high SCI. Harnessing this AMPK-Nrf2 signaling pathway could improve the antioxidant response and help in spinal rewiring to these deafferented phrenic motoneurons to improve diaphragm activity in patients suffering high SCI.

## 1. Introduction

High spinal cord injuries (SCI) lead to the axotomy of descending neuronal pathways within the spinal cord. This results in a permanent loss of motor function, including respiratory insufficiency [1]. For patients living with such injuries, a mechanical ventilation is necessary for survival, and only a few of them may be weaned of ventilatory assistance over time. This spontaneous plasticity is, however, extremely limited. The use of preclinical models is consequently required to decipher the inflammatory and plasticity processes involved. The rodent cervical C2 lateral hemi-section (C2HS) model is the most widely used to evaluate the impact of high SCI on spinal neuroplasticity and neuroinflammation related to the phrenic system [2,3,4,5,6,7,8,9].

More precisely, descending respiratory pathways originate from the rostral-ventral respiratory group (rVRG) located in the rostro-ventral medulla [10]. They connect monosynaptically and bilaterally to the phrenic motoneurons located between C3 and C6 spinal segments of the spinal cord in several species, such as human [11], cat [12,13], rat [14], and mouse [15]. The rVRG also indirectly connects phrenic motoneurons via phrenic interneurons [16,17,18]. Phrenic motoneurons innervate the diaphragm (main inspiratory muscle) bilaterally through the phrenic nerves. In this preclinical model, a unilateral C2HS induces a disruption of the bulbospinal tract from one side of the brainstem to the phrenic motoneurons innervating the diaphragm, leading to hemi-diaphragm paralysis. The contralateral side of the C2 hemi-injury remains intact, allowing the survival and recovery of the animal. This also leads to inflammatory processes at the site of injury [19,20], but also in the aera of deafferented motoneurons [9,21].

Glutamate excitotoxicity participates in these inflammatory processes. SCI induce the axotomy of glutamatergic neurons innerving motoneurons, which leads to excessive release of glutamate [22]. This glutamate then binds to its α-amino-3-hydroxy-5-methyl-4-isoxazolepropionic acid (AMPA), Kainate, and N-methyl-D-aspartate (NMDA) receptors. These ionotropic receptors are permeable to monovalent cations (sodium and potassium), as well as to calcium for NMDA and for certain AMPA receptors. An overactivation of these ionotropic receptors leads to a strong influx of calcium into the postsynaptic neuron, resulting, among other things, in oxidative stress [23]. More precisely, this leads to an increase in the production of reactive oxygen species (ROS) by the mitochondria and their accumulation in the cell body [24]. To cope with this cellular stress [25], there is an increase in the production of antioxidant molecules in the cell. This increase occurs through the modulation of AMPK [26]. AMPK is activated in case of increased ATP concentration in the cytoplasm, which happens in case of increased activity of the electron transport chain in the mitochondrion due to ROS formation [24]. AMPK activation is performed through its phosphorylation by LKB1. AMPK can also be activated by the increase in calcium concentration in the cytoplasm [27]. Phosphorylated AMPK (pAMPK) is then involved in transcription factors’ phosphorylation, facilitating their translocation to the nucleus where they induce the expression of diverse genes involved in metabolism and antioxidative processes [27], such as nuclear factor erythroid 2-related factor (Nrf2) [28].

Nrf2 is a key regulator of the cellular anti-inflammatory and antioxidant responses [29,30,31]. Nrf2 is sequestered in the cytoplasm by Kelch-like ECH-associated protein 1 (Keap1) through the Neh2 domain and is degraded through ubiquitin-dependent proteasomal degradation under physiological conditions [32,33]. In case of cellular stress, Nrf2 is released, and then translocates into the nucleus where it binds to antioxidant response elements (ARE), leading to the transcription of related genes coding for anti-inflammatory, antioxidant, and cytoprotective proteins, such as NADPH quinine oxidoreductase 1 (NQO1) and heme oxygenase 1 (HO-1) [34,35]. Following SCI, it has been shown that in mice, a reduction of Nrf2 expression [36] or its absence (Nrf2 knockout mice) [37] leads to an increase in neuronal death associated with locomotor dysfunction [36,37]. Nrf2 indeed has an important role in the reduction of TNF-α and MMP9 production following SCI [38], with these molecules being involved in spinal neuroinflammation [39,40]. Activation of Nrf2 has therefore been considered as a potential treatment following SCI, considering its anti-inflammatory and antioxidant effects [29,30,31]. The administration of Nrf2 activators has shown beneficial effects in this context, such as a reduction in inflammatory cytokine production, and an increase in Nrf2 expression and its related signaling pathway in neuronal and glial cells [29]. These results display the pivotal role of Nrf2 in neuroinflammation and the resulting functional outcome following SCI.

A better understanding of the role of Nrf2 and its related pathway on inflammation and oxidative stress in this context could help in harnessing its beneficial effects to develop new effective strategies aiming to reduce neuroinflammation and improve the functional outcome. To date, however, no study has evaluated the oxidative stress and Nrf2-mediated antioxidant and anti-inflammatory activities at the level of phrenic motoneurons following cervical SCI. We therefore investigated these inflammatory-related processes in phrenic motoneurons at different time points post-injury (P.I.) in rats following a C2 spinal cord lateral hemi-section.

## 2. Materials and Methods

### 2.1. Ethics Statement

All experiments reported in this manuscript conformed to policies laid out by the National Institutes of Health (Bethesda, MD, USA) in the Guide for the Care and Use of Laboratory Animals and the European Communities Council Directive of 22 September 2010 (2010/63/EU, 74) regarding animal experimentation (Apafis #2017111516297308_v3). These experiments were performed on 3–4-month-old male Sprague–Dawley rats (Janvier, France). The animals were dual-housed in individually ventilated cages in a state-of-the-art animal care facility (2CARE animal facility, accreditation A78-322-3, France), with access to food and water ad libitum with a 12 h light/dark cycle. The Ethics Committee of the RBUCE-UP chair of Excellence (University of Paris-Sud, grant agreement No. 246556) and the University of Versailles Saint-Quentin-en-Yvelines approved these experiments.

### 2.2. Animal Groups and Surgical Preparation

In this study, 63 animals were used, divided into 9 groups: control (*n* = 4), Sham animals (at 1 h (*n* = 6), 1 day (*n* = 7), 3 days (*n* = 7), and 7 days (*n* = 7) post-surgery), and C2-injured animals (at 1 h (*n* = 8), 1 day (*n* = 8), 3 days (*n* = 8), and 7 days (*n* = 8) post-injury). None of the C2-injured animals died during or after the procedure, however one Sham animal died at 1 h due to a technical issue. All animals (except *n* = 4 of each group) received an intrapleural injection with the retrograde tracer Cholera toxin B fragment (CTB, List Biologicals, Campbell, CA, USA) to identify phrenic motor neurons in the cervical spinal cord, similar to that described previously [41,42,43,44]. Briefly, each rat was anaesthetized with isoflurane (Iso-vet, Piramal, Voorschoten, Netherlands; ~1.5% in O_2_), placed in the supine position on a surgical table while spontaneously breathing into a face mask. The lateral sides of the rib cage were shaved and gently palpated to identify the fifth intercostal space at the anterior axillary line. A sterilized 26G needle connected to a 50 μL Hamilton syringe was used to inject 25 μL of a 0.2% CTB (dissolved in sterile injectable saline) between the 5th and the 6th ribs into the pleural space on each side. The needle was modified so that it could be inserted no further than 6 mm below the surface of the skin. After the injections, the animals were monitored closely for signs of respiratory distress associated with an eventual pneumothorax. Isoflurane was turned off and the rats were monitored for 30 min. None of the rats showed signs of respiratory distress. Three days after CTB injection, rats underwent C2 or sham surgery. As described previously [7], buprenorphine (Buprécare, 50 μg/kg, analgesic, Axience, Pantin, France), carprofen (5 mg/kg, anti-inflammatory, Rymadyl, Zoetis, Malakoff, France), enrofloxacin (4 mg/kg, antibiotics, Baytril, Bayer, Leverkusen, Germany), and medetomidine (0.1 mg/kg, Médétor, Virbac, Carros, France) were administered subcutaneously 10–20 min before inducing isoflurane anesthesia in a closed chamber (in 100% O_2_). Rats were orotracheally intubated and ventilated with a rodent ventilator (model 683; Harvard Apparatus, South Natick, MA, USA). Anesthesia was maintained throughout the procedure (1.5–2% isoflurane in 100% O_2_). Skin and muscles were retracted, and a C2 laminectomy and durotomy were performed. For the C2-injured animals, the spinal cord was sectioned unilaterally just caudal to the C2 dorsal roots with micro-scissors, followed with a micro-scalpel to ensure the section of all the remaining fibers, as described previously by our team [7]. The wounds and skin were sutured shut. Sham rats underwent the same procedures without hemi-section. An intra-muscular injection of atipamezole (0.5 mg/kg, Revertor, Virbac, Carros, France) was administered to reverse medetomidine. The isoflurane was discontinued, the endotracheal tube was pulled off, and the rats were monitored throughout recovery. All animals received an injection of analgesic (tramadol, 15 mg/kg), antibiotics (enrofloxacin, 4 mg/kg), and anti-inflammatory (carpofen, 5 mg/kg) drugs for 2 days post-surgery.

### 2.3. Tissue Harvesting and Sample Processing

After the proper time post-surgery (1 h, 1 day, 3 days, or 7 days), the animals were deeply anesthetized (5% isoflurane) and a lethal injection of 1 mL of urethane (0.2 g/mL, i.c., Sigma-Aldrich, Darmstadt, Germany) was performed. Then, the animals were transcardially perfused with cold (4 °C) heparinized saline (5 UI/mL) followed by Antigenfix solution (DIAPATH, Martinengo (BG), Italy). The C1–C6 spinal cords were dissected out and stored in Antigenfix solution overnight at 4 °C, then transferred to an ascending sucrose concentration (20% to 30% sucrose in phosphate-buffered saline (PBS) 1X (BP665-1; Fisher Scientific, Illkirch, France) until they lost buoyancy. The cervical spinal cord was divided into two distinct parts. The C1–C3 spinal cord was sectioned longitudinally (30 µm thickness) with a cryostat (NX70, Thermo Fisher Scientific, Waltham, MA, USA), mounted on slides, and the injury was assessed with cresyl violet staining: 10 min in cresyl violet solution (0.001% cresyl violet acetate (C5042-10G, Sigma-Aldrich) and 0.125% glacial acetic acid (A/0400/PB15, Fisher Scientific) in distilled water), 1 min in 70% ethanol, 1 min in 95% ethanol, 2 × 1 min in 100% ethanol (E/0600DF/17, Fisher Scientific), and 2 min in xylene (X/0100/PB17, Fischer Scientific), and cover-slipped with Eukitt^®^ mounting medium. A brightfield microscopy (Aperio AT2, Leica, Nanterre, France) was used to reconstruct the extent of the injury. Each injury was reported on a stereotaxic C2 metameric transverse plane, as described previously [6,7]. Each reconstructed injury was digitized and analyzed with ImageJ 1.53n software (National Institutes of Health, Bethesda, MD, USA). The percentage of each injured side was evaluated by reference to a complete hemi-section (which is 100%, as previously described [7,45]) (Figure 1).

### 2.4. Immunohistochemistry and In Situ Enzymatic Reaction

The C3–C6 spinal cord was transversally cut (30 µm thickness) with a cryostat, and the sections were stored in a cryoprotectant solution (sucrose 30% (Pharmagrade, 141621, AppliChem, Darmstadt, Germany), ethylene glycol 30% (BP230-4, Fisher Scientific), and polyvinylpyrrolidone 40 (PVP40-100G; Sigma-Aldrich 1% in phosphate-buffered saline PBS 1X)) at −22 °C for further investigations. Sections were prepared for double-labeling with CTB and specific antibodies incubations for: CD11b (1/250, CBL1512, Merck Millipore, Guyancourt, France), CD68 (1/300, MAB1435, Millipore-Merck), iNOS (1/500, AB5382, Merck Millipore), p-AMPK (1/200, 09–290, Merck Millipore), or Nrf2 (1/500, ab31163, Abcam, Cambridge, UK). For dual-labeling with CTB and the molecule of interest, free floating sections were first washed with PBS 0.1 M, incubated in a custom-made blocking solution (5% normal donkey serum, 0.1% triton in PBS 1X for 30 min), and incubated overnight at 4 °C with goat anti-CTB antibody (1/5000, 227040, Calbiochem, Millipore-Merck) and one antibody at the concentration described above. After 3 × 5 min washes, the sections were incubated with a secondary antibody: Alexa Fluor 594 donkey anti-goat (1/1000, A11058, Invitrogen, Waltham, MA, USA), Alexa Fluor 488 donkey anti-rabbit (A21206), or anti-mouse (A21202), for the other antibodies (1/2000, Molecular probes, Eugene, OR, USA) at room temperature for 2 h and 30 min. The sections were then washed several times in PBS 1× and mounted on slides using an anti-fade solution (Prolong Gold antifade reagent, P36930, Invitrogen). Negative controls in which primary and/or secondary antibodies were excluded from the incubation period were also performed. Images of the sections were taken with a Hamamatsu ORCA-R² camera mounted on an Olympus IX83 P2ZF microscope (Tokyo, Japan). Images were analyzed using ImageJ 1.53n software (National Institutes of Health).

For NADPH diaphorase (Nadph-Ox.) histochemistry, free floating sections were incubated in 0.1 M PBS containing 0.2% triton X-100, 1 mg/mL of β-nicotinamide adenine dinucleotide phosphate (β-NADPH, N-1630, Sigma-Aldrich, Darmstadt, Germany), and 1 mg/mL of nitro blue tetrazolium (N-5514, Sigma-Aldrich) during 30 min at 37 °C. Then, CTB immunohistochemistry was performed for identifying the phrenic motoneurons, and the sections were mounted on slides, as previously described. A brightfield microscopy (Aperio AT2, Leica) was used to capture images of the different sections stained with NADPH oxidase.

### 2.5. Protein Extraction and Western Blotting

After the proper post-lesion time (1 h, 1 day, 3 days, or 7 days, *n* = 4 for each group, including *n* = 4 at each time post-surgery for the uninjured group), animals were euthanized (5% isoflurane in 100% O_2_ followed by 1 mL i.c. of urethane, U2500-100 g, Sigma, 0.2 g/mL) and fresh segments of the spinal cord (C3–C6) were collected and frozen immediately in liquid nitrogen. The C1–C3 area of each animal was also collected, and the extent of the injury was verified as described previously. For immunoblot analysis, flash-frozen fresh spinal segments (C3–C6) were lysed with cold RIPA buffer (150 mM of NaCl, 1% Triton X-100, 0.5% sodium deoxycholate, 0.1% SDS, 50 mM of Tris-HCl, pH 7.5, supplemented with a protease inhibitor mixture (Roche Diagnostics, Indianapolis, IN, USA)). Protein concentrations were determined with the DC protein assay kit (5000111EDU, BioRad, Hercules, CA, USA). Twenty µg of total proteins was resolved by SDS-PAGE (4–20% precast gels, BioRad) and transferred to polyvinylidene difluoride membrane (Immobilon-FL, Merck). Membrane blocking was performed in 5% Milk/TBST (10 mM of Tris-HCl, pH 7.4, 150 mM of NaCl, and 0.1% Tween 20) for 1 h prior to incubation with primary antibodies. Primaries antibodies were for Nrf2 (1/1000, sc-13032, Santa-Cruz, Dallas, TX, USA), HO-1 (1/1000, sc-10791, Santa-Cruz), NQO1 (1/1000, AB34176, Abcam), and GAPDH (1/5000, CB1001, Merck Millipore). The corresponding IRDye680- or IRDye800-conjugated secondary antibodies (LI-COR, Bad Homburg vor der Höhe, Germany) were used at dilution ± 1/4000. Immunoreactivity was visualized using the Odyssey Imaging system and the Image Studio software (LI-COR).

### 2.6. Data Analyses and Statistics

All the data were presented as mean ± SD and statistics were considered significant when *p* < 0.05. For immunolabeling analysis, the mean of each motoneuron’s values was considered for each animal group. One-way analysis of variance (ANOVA) or Kruskal–Wallis (Fisher LSD method) one-way ANOVA on ranks (when the normality test or variance test failed, Dunn’s method) were used to compare the values between the different groups. SigmaPlot 12.5 software (Systat Software, San Jose, CA, USA) was used for all analyses.

## 3. Results

### 3.1. Inflammatory Processes in Phrenic Motoneurons

The C2 spinal cord injury leads to the activation of immune cells in the area of phrenic motoneurons, characterized by an increase in the occupied surface by CD11b+ and CD68+ cells until at least 7 days (d) P.I. (Appendix A). This corresponds to previous observations performed following spinal cord injury [21,46].

In phrenic motoneurons specifically, expression of NADPH oxidase, an oxidative stress marker, has been evaluated (Figure 2A). This expression tended to increase in phrenic motoneurons over time until it became significant compared to the uninjured group (36.94 ± 6.32 AU) at 3 d P.I. for the intact side (54.84 ± 3.15 AU, *p* < 0.05). This increase was further amplified at 7 d P.I. for the intact side (83.78 ± 2.98 AU) when compared to the uninjured group and 1 h P.I. (43.18 ± 2.28 AU, *p* < 0.05). For the injured side, the values were significantly increased at 7 d P.I. (79.06 ± 4.63 AU) when compared to the uninjured group (*p* < 0.05), and to 1 h P.I. groups (46.30 ± 2.36 AU, *p* < 0.05) (Figure 2B).

In the same way, an increase in the expression of iNOS (Figure 3A), another inflammatory-related enzyme, was observed from 1 d P.I. for the intact side (32.46 ± 7.87 AU) and the injured side (31.90 ± 5.81 AU) compared to uninjured animals (18.83 ± 1.73 AU, *p* < 0.05). This increase remained until at least 7 d P.I. for the intact side (35.46 ± 6.92 AU, *p* < 0.05 vs. uninjured) and the injured side (37.86 ± 6.76 AU, *p* < 0.05 vs. uninjured) (Figure 3B).

### 3.2. Effect of SCI on AMPK Phosphorylation

The expression of phosphorylated AMPK (p-AMPK) was then observed in phrenic motoneurons as a marker of the response to cellular stress following SCI (Figure 4A).

At 1 h P.I., this expression was significantly reduced for the intact side (40.00 ± 9.32 AU) and the injured side (39.94 ± 9.44 AU) compared to the uninjured group (59.84 ± 13.90 AU, *p* < 0.001). There was, however, a rebound at 1 d P.I. for the injured side (67.02 ± 12.40 AU) compared to 1 h P.I. (*p* < 0.001). p-AMPK expression remained similar to the uninjured group for both sides at later time points (Figure 4B).

### 3.3. Impact on Nrf2 Expression in Phrenic Motoneurons

The expression of total Nrf2 in phrenic motoneurons (Figure 5A), as well as the percentage of phrenic motoneurons expressing nuclear Nrf2 (Figure 6A), were then evaluated to determine the antioxidant and anti-inflammatory response in these cells.

A significant decrease in total Nrf2 was observed at 1 h P.I. in the intact side (19.79 ± 5.16 AU) and the injured side (18.64 ± 5.35 AU) compared to that of uninjured rats (25.08 ± 6.99 AU, *p* < 0.05). This expression was then restored at 1 d P.I. for both sides (intact side: 25.55 ± 5.95 AU; injured side: 24.95 ± 6.86 AU) compared to 1 h P.I. (*p* < 0.05). This level of expression was then maintained until at least 7 d P.I. for the intact (29.60 ± 5.90 AU, *p* < 0.05 compared to 1 h P.I.) and the injured (28.44 ± 6.21 AU, *p* < 0.05 compared to 1 h P.I.) sides (Figure 5B).

Concerning the percentage of phrenic motoneurons expressing nuclear Nrf2, a significant increase was observed at the injured side at 1 d P.I. (0.59 ± 0.13%) compared to 1 h P.I. (0.05 ± 0.05%, *p* < 0.05). This increased expression persisted until at least 7 d P.I. (0.54 ± 0.10, *p* < 0.05 compared to 1 h P.I.) (Figure 6B). Note, however, that there was a decreasing trend at 1 h P.I. for both sides compared to the uninjured group, and for the intact side, the tendency resembled the injured side (Figure 6B).

### 3.4. Impact on Nrf2 Signaling Pathway in the C3–C6 Spinal Cord

Finally, quantification of Nrf2 and two of its effectors, HO-1 and NQO1, was carried out on the complete C3–C6 spinal cord to evaluate the activation of the Nrf2 pathway globally at the level of phrenic motoneurons (Figure 7). Concerning Nrf2 expression, for the injured side, a reduction was observed at 1 h P.I. (0.79 ± 0.20 AU) compared to the uninjured group (1.34 ± 0.18 AU, *p* = 0.045). A rebound was observed at 3 d P.I. (1.75 ± 0.56 AU) compared to 1 h P.I. (*p* < 0.001). Then, there was a significant decreased at 7 d P.I. (1.07 ± 0.15 AU) compared to 3 d P.I. (*p* = 0.016), and a return to values similar to the uninjured ones (*p* > 0.05) (Figure 7A). For the intact side, an increase was observed between 1 h P.I. (1.17 ± 0.37 AU) and 3 d P.I. (2.77 ± 0.18 AU, *p* < 0.001). Interestingly, this increased expression was significantly higher for the intact side than for the injured side (*p* < 0.001). Values then significantly decreased at 7 d P.I. (1.01 ± 0.14 AU, *p* < 0.001, compared to 3 d P.I.) and returned to those seen in the uninjured group (*p* > 0.05) (Figure 7A).

Concerning HO-1 expression on the injured side, a significant decrease was observed at 1 d P.I. (0.71 ± 0.14 AU) compared to uninjured animals (1.04 ± 0.34 AU, *p* = 0.039). Similar to Nrf2 expression, a rebound was observed at 3 d P.I. (2.17 ± 0.35 AU, *p* < 0.001 compared to 1 d P.I.), and for the intact side, a drastic increase was observed at 3 d P.I. (2.28 ± 0.44 AU) compared to 1 d P.I. (0.70 ± 0.19 AU, *p* < 0.001). HO-1 expression then decreased at 7 d P.I. for both sides (injured side: 1.26 ± 0.05 AU; intact side: 0.79 ± 0.09 AU) compared to 3 d P.I. (*p* < 0.001) (Figure 7B).

Finally, NQO1 expression was significantly reduced for the intact side at 1 d P.I. (0.84 ± 0.19 AU) compared to uninjured animals (0.97 ± 0.18 AU, *p* = 0.006) and compared to the injured side (0.51 ± 0.001 AU, *p* = 0.038). A drastic increase was then observed at 3 d P.I. for both sides (intact side: 2.23 ± 0.34 AU; injured side: 1.88 ± 0.39 AU) compared to 1 d P.I. (*p* < 0.001). At 7 d P.I, while the expression remained similar to 3 d P.I. for the injured side (1.63 ± 0.19 AU, *p* > 0.05), there was a significant reduction for the intact side (1.18 ± 0.16 AU) compared to 3 d P.I. (*p* < 0.001) and to the injured side (*p* = 0.008) (Figure 7C).

## 4. Discussion

The present study is the first to evaluate the impact of high SCI on oxidative stress, and AMPK and Nrf2 signaling pathway modulation in phrenic motoneurons in a rat model.

Following SCI, axotomy of descending spinal pathways leads to general neuroinflammation below the lesion site. This is due in great part to the glutamate excitotoxicity phenomenon [23]. We observed an activation and a recruitment of immune cells over time around the identified phrenic motoneurons following C2HS in rats. Our results correlate with other studies showing that the surface occupied by OX42- (microglia/macrophages marker) or GFAP (astrocytes marker)-labeled cells is increased in the C4 spinal cord in the same model [21]. This shows the gradual establishment of neuroinflammatory processes at the level of phrenic motoneurons in the C3–C6 spinal cord. These phrenic motoneurons are also affected by the injury, and successive glutamate excitotoxicity [30]. Here, we showed that expressions of NADPH oxidase and iNOS, two enzymes producing ROS and reactive nitrogen species, respectively, are gradually increased over time in the intact and the injured sides, indicating that phrenic motoneurons’ homeostasis is impacted. Others also pointed out molecular modifications in phrenic motoneurons’ function. In a model of C2HS in rats, a reduction in NMDA receptor mRNA, but not in AMPA receptor mRNA, was observed at 3 d P.I. in phrenic motoneurons [47]. This could be a response to the overactivation of these receptors due to glutamate excitotoxicity. In a model of C2 hemi-contusion in rats, a decrease in the membrane/cytosolic ratio for KCC2 expression was observed for the intact and the injured sides, which reflects a modification in chloride flux in phrenic motoneurons in rat [48]. All these data show that following cervical SCI, deafferented and non-deafferented phrenic motoneurons’ homeostasis is highly modified, however without any impact on their survival [49]. This cellular stress appearing in non-deafferented phrenic motoneurons could be explained by the propagation of inflammatory processes from the injured side, or even by a potential functional adaptation for the loss of activity on the injured side.

Unexpectedly, we observed a significant reduction in pAMPK and total Nrf2 in phrenic motoneurons at 1 h P.I., and a marked tendency for the percentage of phrenic motoneurons expressing nuclear Nrf2. This reduction was, however, only temporary, considering a rebound characterized by a significant increase was observed at 1 d P.I. for these 2 molecules. This phenomenon could be explained by an increased activity of the ubiquitin proteasome system, leading to the degradation of pAMPK and Nrf2. Indeed, a previous study showed that in hippocampal neurons, an upregulation of action potentials leads to an increase in protein degradation via the ubiquitin proteasome system [50]. In our model, glutamate excitotoxicity results in immediate overactivation of phrenic motoneurons, which could lead to a transient increase in ubiquitin proteasome system activity, and subsequent pAMPK and Nrf2 degradation. NADPH oxidase and iNOS were either not expressed or expressed at a very low level physiologically, and the absence of a reduction in their expression at 1 h P.I. was not surprising. However, we are the first to observe such a drastic reduction in metabolism and anti-inflammation/antioxidant-related molecules at 1 h P.I. in phrenic motoneurons.

Phrenic motoneurons have the capacity to respond to the cellular stress caused by glutamate excitotoxicity, involving activation of AMPK and Nrf2 signaling pathways. The reduction in neuron metabolism due to the spinal shock [51] could explain the drastic rebound in expression observed at 3 d following the injury. As AMPK is a key regulator of cellular metabolism [27], its presence is indeed required in case of reduced neuron metabolism. Following SCI, higher AMPK signaling is known to increase autophagy [52] and reduce apoptosis in rats [53,54] and mice [55]. On the contrary, inhibition of AMPK leads to enhanced axonal regeneration following dorsal column crush in mice [56]. This shows the pivotal role of AMPK following SCI and could explain its rebound in expression in phrenic motoneurons at 3 d P.I., considering no study explored AMPK function in phrenic motoneurons in this context. Concerning Nrf2, the cellular stress caused by glutamate excitotoxicity leads to anti-inflammatory and antioxidant responses, which requires Nrf2 signaling [31]. This could explain the rebound also observed for Nrf2 expression using immunolabeling at 1 d P.I., as well as the increased expression of Nrf2, HO-1, and NQO1 observed at 3 d P.I. using immunoblotting. This difference in timing could be explained by the fact that immunoblotting was performed on the total C3–C6 spinal cord, which contains not only phrenic motoneurons but also other neuronal and glial cells. On the contrary, immunolabeling analysis was performed only on phrenic motoneurons in the C3–C6 spinal cord. Considering that other neuronal and glial cells are in higher quantity than phrenic motoneurons, a difference in inflammatory kinetics would erase the increase in Nrf2 expression observed in phrenic motoneurons, specifically when using the immunoblotting method. Nrf2 appeared to be beneficial following injury [30,31]. In mice lacking Nrf2, SCI leads to more severe locomotor dysfunction and neural death compared to control animals [57], whereas activation of the Nrf2 pathway in rats leads to improved locomotor function and neuroprotective effects following SCI [31,37,58]. Nrf2 is also expressed in glial cells, including astrocytes. Hyperactivation of Nrf2 in astrocytes via Keap1 depletion leads to a reduced loss of myelin and oligodendrocytes in mice following spinal cord contusion [59]. These data as well as our results show that Nrf2 pathway activity in spinal neuronal and glial cells participates in the defense response against oxidative stress following spinal cord injury.

Here, we proposed to link glutamate excitotoxicity occurring in phrenic motoneurons following high spinal cord injury [60] with anti-inflammatory and antioxidant responses involving AMPK [52] and Nrf2 signaling [31]. This AMPK-Nrf2 pathway was indeed shown to be associated with neuroprotection [61]. Axotomy in the upper spinal cord (C2) led to glutamate excitotoxicity in phrenic motoneurons located in the lower spinal cord (C3–C6). This induced overactivation of glutamatergic receptors and high calcium influx [60]. It resulted in high production of ROS [62] and activation of AMPK via phosphorylation due to a higher ATP concentration [27]. pAMPK then facilitated Nrf2 nuclear translocation [28]. This allowed transcription of antioxidant and anti-inflammatory molecules (Figure 8).

## 5. Conclusions

Taken together, our results illustrated the implication of the AMPK-Nrf2 pathway in phrenic motoneurons’ response to oxidative stress following SCI. This improved our current knowledge about phrenic motoneurons’ function and paves the way for future therapies aiming to improve phrenic motoneurons’ response to oxidative stress and neuroinflammation. Deciphering the time-dependent changes in AMPK-Nrf2 signaling and its eventual pharmacological modulation could indeed help in spinal rewiring of these deafferented phrenic motoneurons and could lead to improved diaphragm activity in patients suffering high SCI.

## Figures and Tables

**Figure 1 antioxidants-11-01665-f001:**
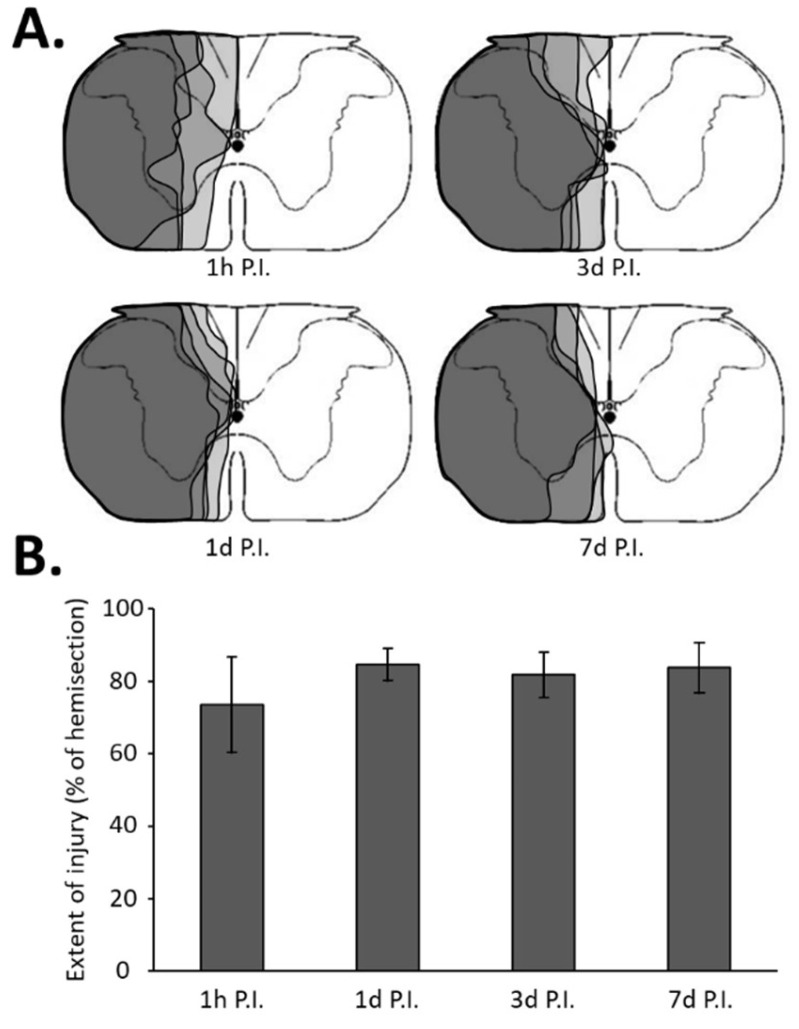
Extent of injury following a C2 spinal cord hemi-section. (**A**) Representation of a transversal view of the C2 spinal cord and representative extent of injury for each group in gray, i.e., at 1 h post-injury (P.I.), 1 day P.I., 3 days P.I., and 7 days P.I. (**B**) Extent of injury quantification in percentage compared with 100% uninjured hemi-spinal cord. The quantification has been made only in the ventral part where the phrenic motoneurons are located. There was no difference between the different groups (one-way ANOVA, *p* = 0.277).

**Figure 2 antioxidants-11-01665-f002:**
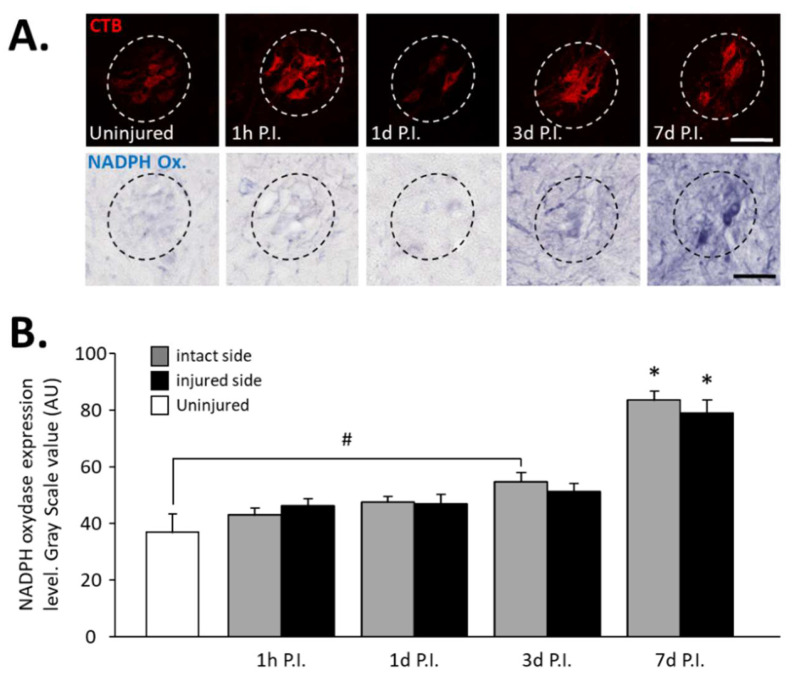
NADPH oxidase expression in phrenic motoneurons following C2 hemi-section. (**A**) Representative images showing expression of NADPH oxidase in phrenic motoneurons labeled with CTB in uninjured animals and in denervated phrenic motoneurons in C2 hemisected rats, 1 hour (h), 1 day (d), 3 d, and 7 d post-injury (P.I.). (**B**) Quantification of NADPH oxidase in phrenic motoneurons for uninjured animals, and intact and injured sides of C2 hemisected animals 1 h, 1 d, 3 d, and 7 d following injury. # 3 d P.I. intact side, compared with uninjured group, *p* < 0.05. * 7 d P.I. compared to uninjured group and 1 h P.I. corresponding side, *p* < 0.05.

**Figure 3 antioxidants-11-01665-f003:**
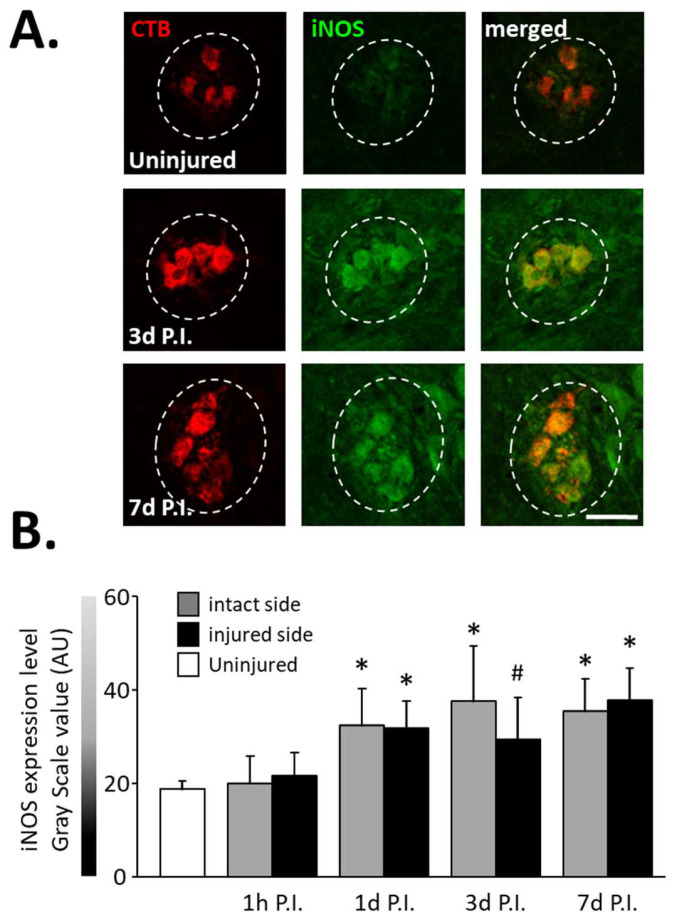
iNOS expression in phrenic motoneurons following C2 hemi-section. (**A**) Representative images showing expression of iNOS in phrenic motoneurons labeled with CTB in uninjured animals and in denervated phrenic motoneurons in C2 hemisected rats, 3 and 7 days post-injury (P.I.). (**B**) Quantification of iNOS in phrenic motoneurons for uninjured animals, and intact and injured sides of C2 hemisected animals 1 h, 1 d, 3 d, and 7 d following injury. # 3 d P.I. injured side, compared with uninjured group, *p* < 0.05. * Compared to uninjured group and 1 h P.I. corresponding side, *p* < 0.05.

**Figure 4 antioxidants-11-01665-f004:**
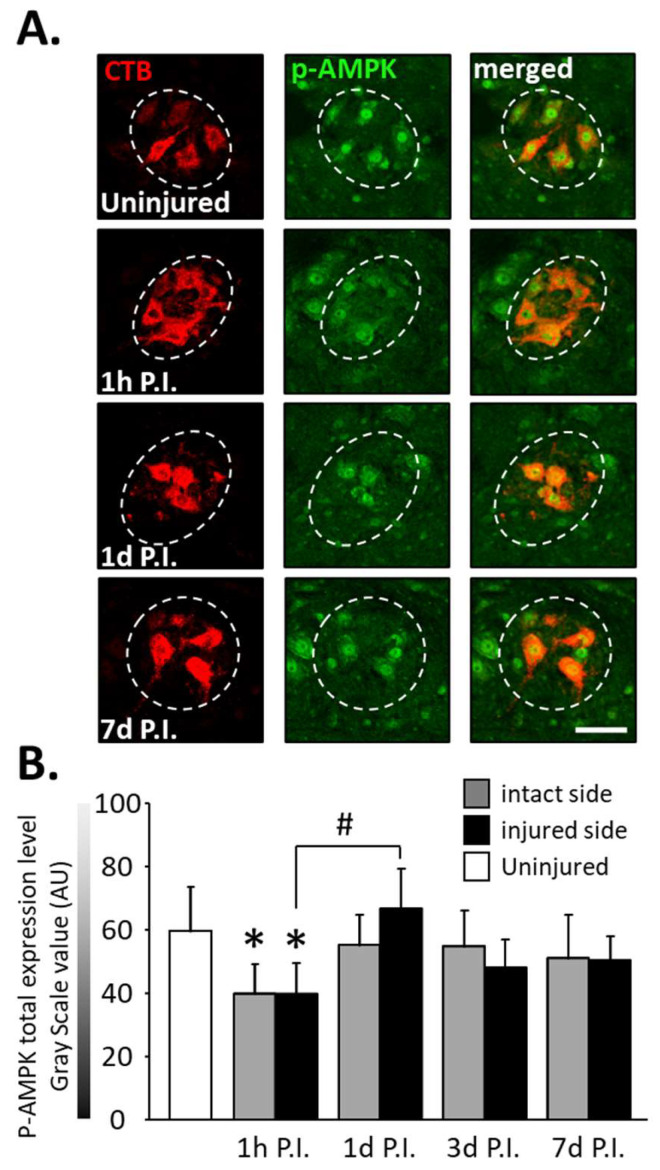
Phosphorylated AMPK (pAMPK) expression in phrenic motoneurons following C2 hemi-section. (**A**) Representative images showing the expression of pAMPK in phrenic motoneurons labeled with CTB in uninjured animals and in denervated phrenic motoneurons in C2 hemisected rats, 1 h, 1 d, and 7 d post-injury (P.I.). (**B**) Quantification of pAMPK in phrenic motoneurons for uninjured animals, and intact and injured sides of C2 hemisected animals 1 h, 1 d, 3 d, and 7 d following injury. # 1 d P.I. compared to 1 h P.I. for the injured side, *p* < 0.001. * Compared to the uninjured group, *p* < 0.001.

**Figure 5 antioxidants-11-01665-f005:**
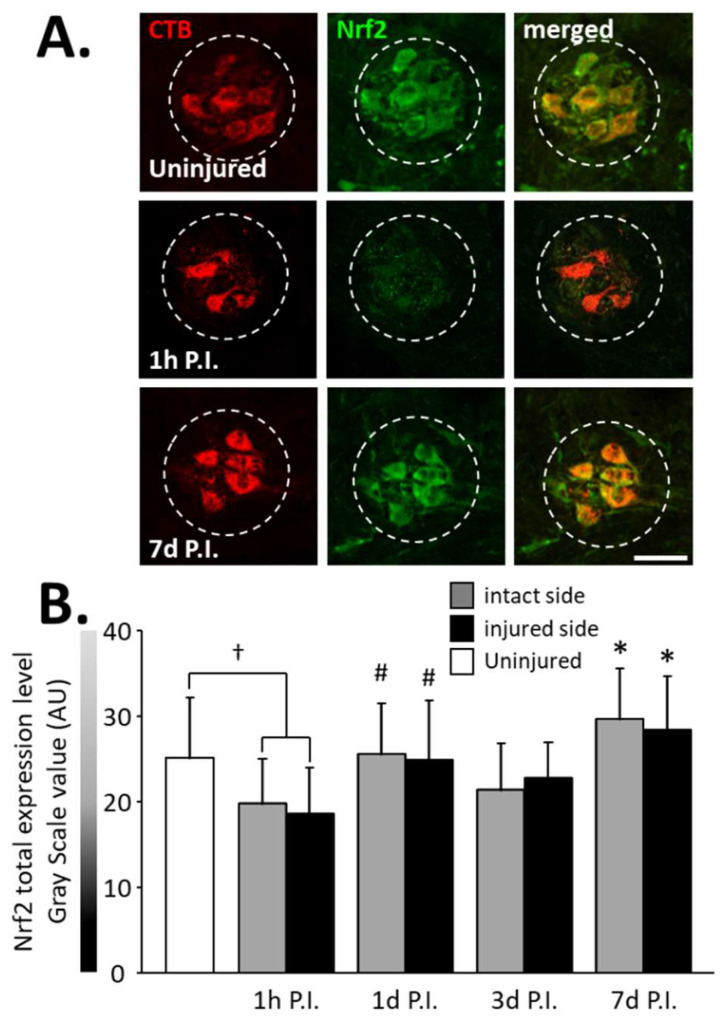
Nrf2 expression in phrenic motoneurons following C2 hemi-section. (**A**) Representative images showing expression of Nrf2 in phrenic motoneurons labeled with CTB in uninjured animals and in denervated phrenic motoneurons in C2 hemisected rats, 1 h and 7 d post-injury (P.I.). (**B**) Quantification of Nrf2 in phrenic motoneurons for uninjured animals, and intact and injured sides of C2 hemisected animals 1 h, 1 d, 3 d, and 7 d following injury. # Compared to 1 h P.I. group, *p* < 0.05. * Compared to uninjured group and 1 h P.I. corresponding side, *p* < 0.05. † Uninjured compared to 1 h intact side and injured side, *p* < 0.05.

**Figure 6 antioxidants-11-01665-f006:**
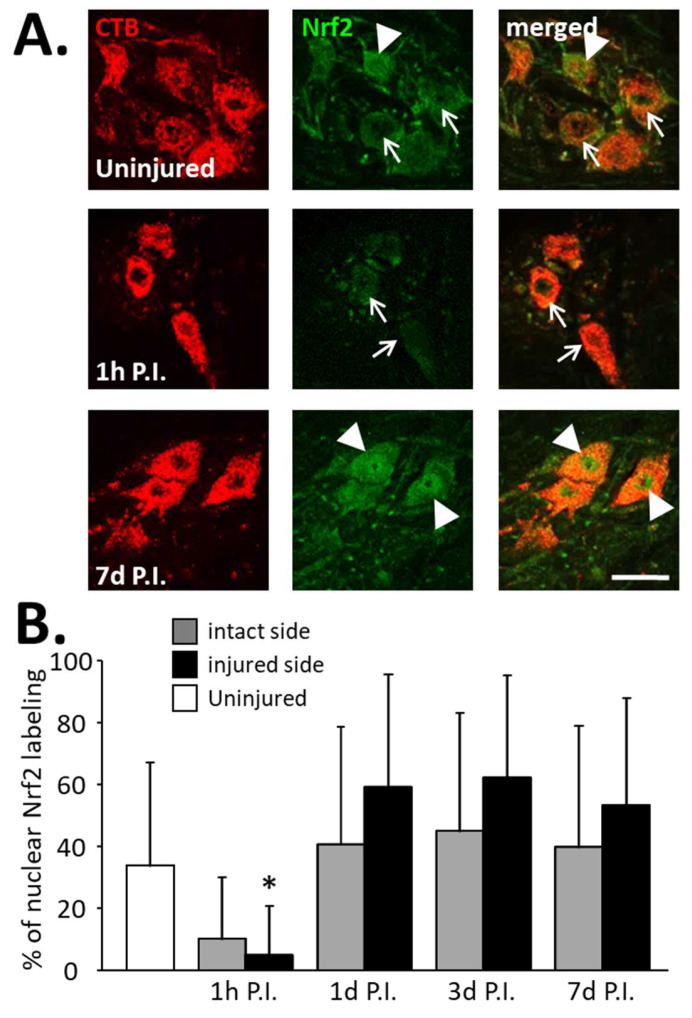
Percentage of phrenic motoneurons expressing nuclear Nrf2 following C2 hemi-section. (**A**) Representative images showing the expression of Nrf2 in phrenic motoneurons labeled with CTB in uninjured animals and in denervated phrenic motoneurons in C2 hemisected rats, 1 h and 7 d post-injury (P.I.). White arrowheads show phrenic motoneurons without nuclear Nrf2 expression. White arrows show phrenic motoneurons with nuclear Nrf2 expression. (**B**) Percentage of phrenic motoneurons expressing nuclear Nrf2 for uninjured animals, and intact and injured sides of C2 hemisected animals 1 h, 1 d, 3 d, and 7 d following injury. * Compared to 1 h P.I., 3 d P.I., and 7 d P.I. corresponding sides, *p* < 0.05.

**Figure 7 antioxidants-11-01665-f007:**
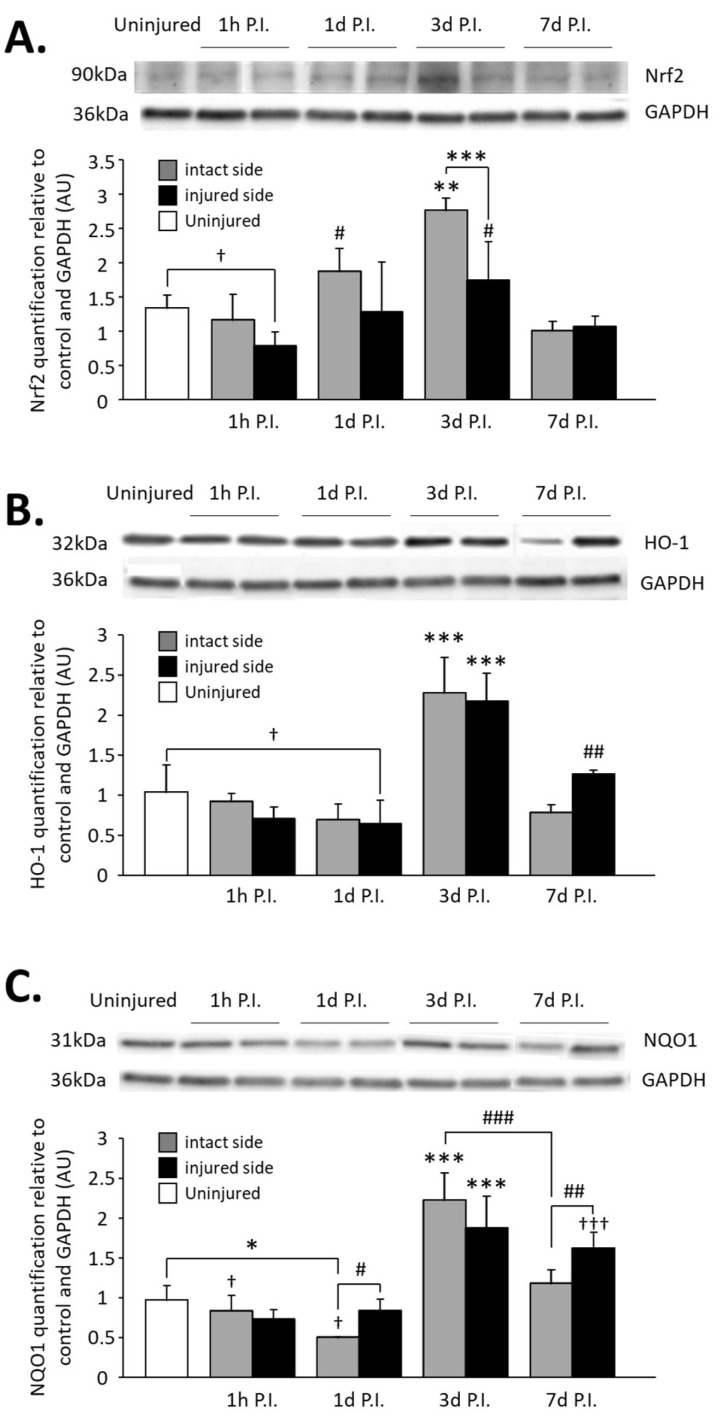
Quantification of Nrf2 and two of its effectors, HO-1 and NQO1, in the C3–C6 spinal cord following C2 hemi-section. (**A**) Representative blot and associated quantification of Nrf2 expression in the C3–C6 spinal cord, 1 h, 1 d, 3 d, and 7 d post-injury (P.I.). ** Compared to uninjured and 1 h, 1 d, and 7 d P.I. corresponding sides, *p* < 0.05. *** 3 d P.I. intact side compared to 3 d P.I. injured side, *p* < 0.001. † Uninjured compared to 1 h P.I. injured side, *p* < 0.05. # Compared to 1 h and 7 d P.I. corresponding side, *p* < 0.05. (**B**) Representative blot and associated quantification of HO-1 expression in the C3–C6 spinal cord, 1 h, 1 d, 3 d, and 7 d post-injury (P.I.). *** Compared to uninjured and 1 h, 1 d, and 7 d P.I. corresponding sides, *p* < 0.001. ## Compared to 1 h and 1 d P.I. corresponding side, *p* < 0.01. † Uninjured compared to 1 d P.I. injured side, *p* = 0.039. (**C**) Representative blot and associated quantification of NQO1 expression in the C3–C6 spinal cord, 1 h, 1 d, 3 d, and 7 d post-injury (P.I.). *** Compared to uninjured and 1 h, 1 d, and 7 d P.I. corresponding sides, *p* < 0.001. ### 3 d P.I. intact side compared to 7 d P.I. intact side, *p* < 0.001. ††† Compared to uninjured and 1 h and 1 d P.I. corresponding sides, *p* < 0.001. ## 7 d P.I. intact side compared to 7 d P.I. injured side, *p* < 0.01. # 1 d P.I. intact side compared to 1 d P.I. injured side, *p* < 0.05. * 1 h P.I. intact side compared to 1 h P.I. injured side, *p* < 0.05. † Compared to 7 d P.I. corresponding side, *p* < 0.05.

**Figure 8 antioxidants-11-01665-f008:**
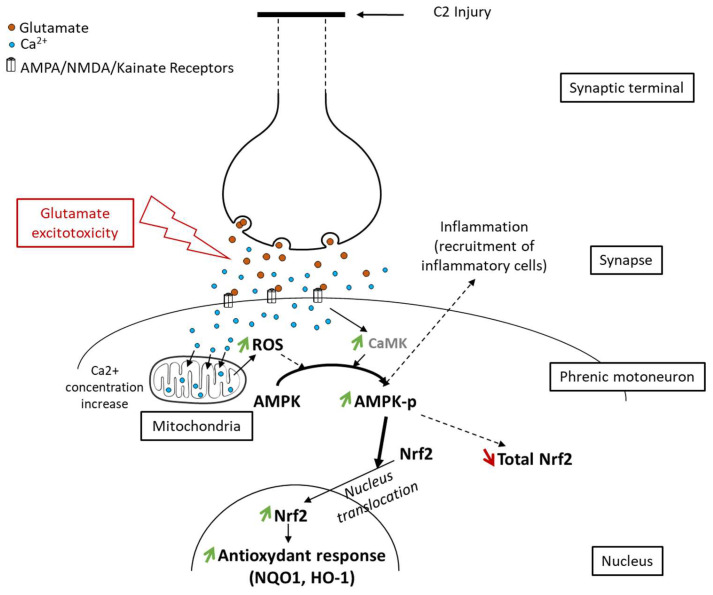
Proposed schematic of Nrf2 signaling pathway activation through phosphorylation of AMPK following spinal cord injury. Following axotomy, synaptic terminals release all their neurotransmitters, including glutamate. This glutamate excessively activates its receptors (AMPA, NMDA, and Kainate), which leads to glutamate excitotoxicity in postsynaptic neurons (here, phrenic motoneurons). Intracellularly, it induces an increase in the calcium concentration, leading to increased reactive oxygen species (ROS) production by mitochondria. These ROS, associated with the increased calcium concentration, provoke AMPK phosphorylation. pAMPK then facilitates Nrf2 nucleus translocation, leading to an increase in the antioxidant response mediated by Nrf2 and its effectors (NQO1, HO-1).

## Data Availability

The data are contained within the manuscript and Appendix A.

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
