# Peer review of "AMPK-Nrf2 Signaling Pathway in Phrenic Motoneurons following Cervical Spinal Cord Injury"

_antioxidants, 2022, doi:10.3390/antiox11091665_

Round 1

Reviewer 1 Report

Comments on Manuscript antioxidants-1862945

Title                AMPK-Nrf2 signaling pathway in phrenic motoneurons following cervical spinal cord injury

Authors          Pauline Michel-Flutot et al

Comments

The author group has nicely designed the experiment to conduct the study of AMPK-Nrf2 signaling pathway in phrenic motoneurons at post injured spinal cord injury. In order to strength the scientific outcomes the author need to address following points.

1.     In figure 1; the authors need to provide the uninjured C2 spinal cord hemisection for comparison of the post-injured stage.

2.     Did the authors found any mortality in any group of animals during the entire experiential timeline.

3.     Did the authors observed any kind of ARDS (acute respiratory distress syndrome) during the experimental time

4.     How the author would confirm that there was no impact of the administration of the following drugs, analgesic (tramadol, 15 mg/kg), antibiotics (enrofloxacin, 4 mg/kg) and anti-inflammatory (carpofen, 5 mg/kg) drugs on the results of the iNOS, pAMPK and Nrf2 expression.

5.     In figure 6, the nuclear Nrf2 expression result value SD in uninjured group of animals is very high please explain. Also explain the sudden fall of nuclear Nrf2 expression in IH PI cross the increasing pattern of its level at 1,3 and 7DPI compared to the uninjured group.

Author Response

The author group has nicely designed the experiment to conduct the study of AMPK-Nrf2 signaling pathway in phrenic motoneurons at post injured spinal cord injury. In order to strength the scientific outcomes the author need to address following points.

We thank the reviewer for its positive comments.

  1. In figure 1; the authors need to provide the uninjured C2 spinal cord hemisection for comparison of the post-injured stage.

The uninjured C2 spinal cord hemisection is represented by the spinal cord itself in white and black in figure 1. It has been clarified in the legend of the Figure 1: “Extent of injury following a C2 spinal cord hemisection: (A) Representation of a transversal view of the C2 spinal cord and representative extent of injury for each group in gray, i.e., at 1h post-injury (P.I.), 1d P.I., 3d P.I. and 7d P.I.; (B) extent of injury quantification in percentage compared with uninjured control hemi-spinal cord injury (SCI) 100%. The quantification has been made only in the ventral part where the phrenic motoneurons are located. There is no difference between the different groups (One Way ANOVA, p = 0.277).”

  1. Did the authors found any mortality in any group of animals during the entire experiential timeline.

Thanks for asking. All animals survived through C2 surgery and during the entire experimental timeline. This C2 hemisection model is widely used for its low mortality rate following injury. We added this sentence in the experimental animal group and surgical preparation: “None of the C2 injured animal died during or after the procedure, except one Sham ani-mal at 1h due to a technical issue.”

  1. Did the authors observed any kind of ARDS (acute respiratory distress syndrome) during the experimental time

No we did not observed any kind of acute respiratory distress syndrome to our knowledge. All animals were monitored several time every day and we did not observed such phenomena, as it has already been published in the literature on this model by our team and others.

  1. How the author would confirm that there was no impact of the administration of the following drugs, analgesic (tramadol, 15 mg/kg), antibiotics (enrofloxacin, 4 mg/kg) and anti-inflammatory (carpofen, 5 mg/kg) drugs on the results of the iNOS, pAMPK and Nrf2 expression.

Thanks for this question. Considering that non-injured animals also received these drugs (as Sham animals), the results/differences observed should be due to the spinal cord injury. It however does not exclude that these drugs could have an effect on the expression of these molecules, but from our statistical analyses so far, the inter-group differences we found are due to the spinal cord injury itself. For ethical purposes, we must use these post-surgery drugs for all groups.

  1. In figure 6, the nuclear Nrf2 expression result value SD in uninjured group of animals is very high please explain. Also explain the sudden fall of nuclear Nrf2 expression in IH PI cross the increasing pattern of its level at 1,3 and 7DPI compared to the uninjured group.

Thanks for the comment. Concerning the high SD in uninjured group, we represented SEM instead of SD for the other groups in Fig 6B. It has been corrected (all values display now the SD for their values). We also double check for the other figures and they display SD.

Concerning the sudden fall of nuclear Nrf2 and the successive increasing pattern (we can call that rebound) of its level at later time points, we hypothesized it could be due to an increased activity of the ubiquitin proteasome system right after injury as detailed in the discussion. This could lead to a reduced expression of Nrf2. Then the “need” for Nrf2 activity due to growing inflammation and oxidative stress could sustain an increase in its expression at later time-points. It will be interesting to test this hypothesis in another study by evaluating the ubiquitin proteasome system activity in specific phrenic motoneurons following spinal cord injury.

Reviewer 2 Report

Antioxidants-1862945

Pauline Michel-Flutot, et al.

AMPK-Nrf2 signaling pathway in phrenic motoneurons following cervical spinal cord injury

Overall Impression:  The overall objective of this study was to determine the antioxidant response in phrenic motoneurons, with emphasis on the AMPK-Nrf2 signaling pathway following C-2 spinal cord lateral hemi-section in a rat model.  The authors showed that there was an abrupt reduction of phosphorylated AMPK and Nrf2 at 1-hour post-injury in phrenic motoneurons.  This was following by a rebound at the 1-day post-injury time point.  In the total spinal core around the phrenic motoneurons the increase phosphorylated AMPK and Nrf2 occurred 3-days post-injury.  This data shows the differential antioxidant response between phrenic motoneurons and other cell types.

Thus, this investigation has generated some novel data that might lead to development of new therapeutic treatment for this type of spinal cord injury.  There are some questions that need to be addressed as well as some changes in the grammar of the English, as detailed below.

1)     There is a need to justify why only male rats were used.  Is there any literature that hints females may respond differently than males.

2)    How were the n (number of animals) chosen for each group?  

3)    Line 263 the phrase compared with the uninjured group appears twice in a row.  Should this be two separate sentences?

4)    Line266 please change has then been observed, to “was observed”.

5)    Line 271 Change expression of p-AMPK remains then similar to ……p-AMPK remains similar to the uninjured group 

6)    Line 452. Change to …..our results illustrate the involvement of the AMP-Nrf2 pathway……

7)    Before moving toward modulating AMPK-Nrf2 signaling in patients with High (meaning upper cervical spine injury) it would be important to show the importance of the time-dependent changes in expression of P-AMPK and Nrf2 in animal models.  Such experiments should be outlined by the authors.

Author Response

Overall Impression:  The overall objective of this study was to determine the antioxidant response in phrenic motoneurons, with emphasis on the AMPK-Nrf2 signaling pathway following C-2 spinal cord lateral hemi-section in a rat model.  The authors showed that there was an abrupt reduction of phosphorylated AMPK and Nrf2 at 1-hour post-injury in phrenic motoneurons.  This was following by a rebound at the 1-day post-injury time point.  In the total spinal core around the phrenic motoneurons the increase phosphorylated AMPK and Nrf2 occurred 3-days post-injury. This data shows the differential antioxidant response between phrenic motoneurons and other cell types.

Thus, this investigation has generated some novel data that might lead to development of new therapeutic treatment for this type of spinal cord injury. There are some questions that need to be addressed as well as some changes in the grammar of the English, as detailed below.

We thank the reviewer for its comments and suggestions. 

1)     There is a need to justify why only male rats were used.  Is there any literature that hints females may respond differently than males.

We choose to use only male to several reasons. Using males allow to avoid hormonal cycle effects in post-traumatic phase since oestrogens have some neuroprotectives effects. There is indeed an existing literature showing a difference in recovery between male and female: Stewart et al. 2021 (DOI: 10.1186/s12974-021-02161-8), McFarlane et al. 2020 (DOI: 10.1089/neu.2019.6931), Farooque et al. 2006 (DOI: 10.1038/sj.sc.3101816) for examples.

 2)    How were the n (number of animals) chosen for each group? 

 The n number of the animal is at the minimum to make sure the statistical power of each test is >0.8, as we used to do. By experience, we already published similar experimentations (C2 hemisection and Immunohistochemistry evaluations, Vinit et al., 2005; 2009; 2011 Darlot et al., 2017 as example) with similar number of animals in each groups which meet the statistical criteria described above.

 3)    Line 263 the phrase compared with the uninjured group appears twice in a row.  Should this be two separate sentences?

Yes there are two separate sentences. The first one gives the description for the # symbol, and the second for the * symbol.

 4)    Line266 please change has then been observed, to “was observed”.

We made the change accordingly.

 5)    Line 271 Change expression of p-AMPK remains then similar to ……p-AMPK remains similar to the uninjured group

We made changes for more clarity. 

6)    Line 452. Change to …..our results illustrate the involvement of the AMP-Nrf2 pathway……

We made the change accordingly

 7)    Before moving toward modulating AMPK-Nrf2 signaling in patients with High (meaning upper cervical spine injury) it would be important to show the importance of the time-dependent changes in expression of P-AMPK and Nrf2 in animal models.  Such experiments should be outlined by the authors.

We rewrite our conclusion accordingly : « Taken together, our results illustrate the implication of the AMPK-Nrf2 pathway in phrenic motoneurons response to oxidative stress following SCI. This improves our current knowledge about phrenic motoneurons function and paves the way for future therapies aiming to improve phrenic motoneurons response to oxidative stress and neuroinflammation. Deciphering the time-dependent changes in AMPK-Nrf2 signaling and its eventual pharmacological modulation could indeed help in spinal rewiring of these deafferented phrenic motoneurons and could lead to improved diaphragm activity in patients suffering high SCI.»